# Identification of novel breast cancer susceptibility loci in meta-analyses conducted among Asian and European descendants

Xiang Shu et al.[#]

Known risk variants explain only a small proportion of breast cancer heritability, particularly in Asian women. To search for additional genetic susceptibility loci for breast cancer, here we perform a meta-analysis of data from genome-wide association studies (GWAS) conducted in Asians (24,206 cases and 24,775 controls) and European descendants (122,977 cases and 105,974 controls). We identified 31 potential novel loci with the lead variant showing an association with breast cancer risk at $P < 5 \times 10^{-8}$. The associations for 10 of these loci were replicated in an independent sample of 16,787 cases and 16,680 controls of Asian women ($P < 0.05$). In addition, we replicated the associations for 78 of the 166 known risk variants at $P < 0.05$ in Asians. These findings improve our understanding of breast cancer genetics and etiology and extend previous findings from studies of European descendants to Asian women.

[#]A full list of authors and their affiliations appears at the end of the paper.

**B**reast cancer is the most commonly diagnosed malignancy and the leading cause of cancer-related deaths in women worldwide[1]. Genetic linkage studies and family-based studies have identified many high- and moderate-penetrance mutations in breast cancer predisposition genes, including *BRCA1, BRCA2, PTEN, ATM, PALB2*, and *CHEK2*[2]. In addition, large-scale genome-wide association studies (GWAS), conducted primarily in Asian and European women, have identified more than 180 susceptibility loci for breast cancer risk[3–8]. These identified loci explain a relatively small proportion of familial relative risk of breast cancer[8].

The Asia Breast Cancer Consortium (ABCC) is the largest breast cancer GWAS consortium conducted in Asian-ancestry populations. We have shown previously that GWAS conducted in Asians could uncover cancer genetic risk variants that are either unique to the Asian population or more difficult to identify in studies conducted in European women[3,4,9–16]. It also has been shown that a large proportion of common susceptibility loci are shared between Asian and European populations, although the lead variants in many loci may differ between these two populations[6,8]. To search for novel breast cancer susceptibility loci, we conducted Asian-specific and cross-ancestry meta-analyses combining the data of the ABCC and the Breast Cancer Association Consortium (BCAC) with a total sample size of approximately 310,000 women (~82,000 Asians and ~228,000 Europeans). We herein report the discovery of 31 potential novel risk loci for breast cancer and the replication of a large number of known breast cancer susceptibility loci in Asian women.

## Results

**Overall associations for newly associated loci.** We identified 28 loci with at least one common variant at each locus showing a significant association with breast cancer risk in the cross-ancestry meta-analysis (i.e., $P < 5 \times 10^{-8}$) (Table 1). None of these lead risk variants reside within a 500 Kb region flanked by any of the 183 previously reported breast cancer risk variants. No obvious inflation in statistical estimates was observed for either Asian-specific or cross-ancestry meta-analysis after excluding known loci (sample size-adjusted $\lambda_{1000} = 1.012$ and 1.001, respectively). No evidence of heterogeneity in associations was observed between the two racial populations except for rs2758598 and rs142360995 (Table 1, $P_{heterogeneity} < 0.05$, consistent in direction). The OR estimates for these 28 SNPs by study within the ABCC and BCAC consortia are presented in Supplementary Data 1 and 2. We explored pleiotropic effects by assessing the newly identified lead variants and their correlated SNPs (in LD with $r^2 > 0.4$ in either Asians or Europeans) from the online catalog of published GWAS (GWAS catalog). Pleiotropy was found for seven of the 28 newly-associated SNPs (Supplementary Table 2).

All of the 28 SNPs showed a nominally significant association ($P < 0.05$) with ER-positive breast cancer risk (Table 2). Fourteen of the 28 risk SNPs were also associated with ER-negative breast cancer risk in the cross-ancestry meta-analysis ($P < 0.05$). Heterogeneity between ER+ and ER- breast cancer risk ($P_{heterogeneity} < 0.05$) was observed for rs73006998, rs7765429, rs144145984, rs78588049, and rs12481286.

Of the 28 SNPs, 22 were investigated in an independent set of 10,829 cases and 10,996 controls included in ABCC and an additional 5958 cases and 5684 controls from studies conducted in Malaysia and Singapore (see Methods). A significant association at $P < 0.05$ was found for 10 SNPs, all with the association direction consistent with our main findings (Supplementary Table 3). Among them, five SNPs showed significant associations at $P < 2.3 \times 10^{-3}$ (0.05/22), including rs3790585 (1p34.1), rs73006998

(3q25.1), rs6940159 (6q27), rs855596 (12q23.2), and rs75004998 (14q24.3).

To uncover possible secondary association signals in newly identified breast cancer susceptibility loci, we performed analyses for SNPs within flanking 500 kb of each lead SNP, with adjustment for the lead SNPs within each dataset. We then conduced meta-analyses to combine the results across studies of Asian women. Six potential secondary associations were identified (conditional $P < 1 \times 10^{-4}$), and all correlated ($r^2 > 0.1$ in 1000 Genome East Asians) except for rs7693779, at 4p12 (Supplementary Table 4).

Of the 28 SNPs newly identified to be associated with breast cancer risk, 13 SNPs are intronic, one in UTR, and 14 in intergenic regions. Using data from ENCODE and Roadmap, we found that the majority of these 28 overlapped with genomic functional biofeatures that were indicative of promoters or enhancers (Supplementary Data 3 and 4). The enrichment analysis supported this observation (Supplementary Fig. 2A). Of particular note is a strikingly strong enrichment signal of transcribed chromatin states that was found for the newly associated loci when compared to all risk loci (Supplementary Fig. 2B). Enrichment signals of multiple histone modifications were also observed for both newly identified and overall association loci (Supplementary Fig. 2C, D). The newly identified loci were enriched particularly for H4K78me2 and H4K20me1. These results indicated that the newly identified loci are tightly involved in active gene transcription events. Of the 28 lead SNPs, four (rs3790585 at 1p34.1, rs6756513 at 2p13.3, rs10820600 at 9q31.1, and rs78588049 at 12q15) intersected with chromosomal segments annotated as strong enhancers or active promoters in breast tissue-originated cell lines. When all SNPs that were in LD with the lead SNPs with $r^2 > 0.8$ in either Asians or Europeans were evaluated, evidence of regulatory function was found for an additional seven (i.e., 1q22-rs2758598, 3q25.1-rs73006998, 3q25.31-rs11281251, 8q22.2-rs2849506, 14q24.3-rs75004998, 15q24.2-rs8027365, and 21q22.3-rs35418111).

**eQTL and gene-based analyses.** To identify target genes of the 28 newly identified lead SNPs, we conducted *cis*-eQTL analyses in four independent datasets in breast tissue. Nine eQTL associations were identified with a $P < 0.05$ with same association direction in two or more independent sets (Supplementary Table 5). Potential candidate genes identified in this analysis included *LINC00886, ybeY metallopeptidase (YBEY), snurportin 1 (SNUPN), mannosidase alpha class 2 C member 1 (MAN2C1), T-Box 1 (TBX1), MutY DNA glycosylase (MUTYH), lysyl oxidase like 2 (LOXL2), stanniocalcin 1 (STC1)*, and *semaphorin 4 A (SEMA4A)*. SNP rs144145984 was the eQTL for both *LOXL2* and *STC1* genes, but the association for *STC1* is much stronger. Similarly, SNP rs8027365 was associated with expression levels of two genes, *MAN2C1* and *SNUPN*.

With the exception of *TBX1* and *LOXL2*, we were able to build breast-tissue and/or cross-tissue models for all other eQTL-identified candidate genes with a prediction $R^2 > 0.01$ (Supplementary Table 6). Expressions of *LINC00886, YBEY, MAN2C1* and *SEMA4A* could be predicted with a high accuracy by both breast tissue and cross tissue models ($R^2 > 0.09$). We imputed expressions of seven genes other than *TBX1* and *LOXL2* and showed that these genes were associated with breast cancer risk in either the ABCC or BCAC data at $P < 0.05$ (Supplementary Table 6). Of these, genes hypothesized to have a tumor-suppressor function included *LINC00886, MAN2C1, SNUPN*, and *STC1*, while *YBEY, SEMA4A*, and *MUTYH* may have an oncogenic role in breast carcinogenesis based on their associations with breast cancer risk (Supplementary Table 7).

**Table 1 Twenty eight novel loci identified by the cross-ancestry meta-analysis.**

| SNP | Chr | BP | Effect | Other | Locus | Asian-specific | | | European-specific | | | Cross-ancestry | | | $I^2_r$, % | $P_{het}$ |
|---|---|---|---|---|---|---|---|---|---|---|---|---|---|---|---|---|
| | | | | | | EAF | OR (95% CI) | P | EAF | OR (95% CI) | P | EAF | OR (95% CI) | P | | |
| rs72906468 | 1 | 177772093 | A | T | 1p36.13 | 0.68 | 1.06 (1.03–1.09) | $1.5 \times 10^{-4}$ | 0.77 | 1.04 (1.02–1.05) | $2.2 \times 10^{-6}$ | 0.76 | 1.04 (1.03–1.05) | $4.0 \times 10^{-9}$ | 0 | 0.59 |
| rs3790585 | 1 | 46023356 | A | T | 1p34.1 | 0.69 | 1.05 (1.02–1.08) | $1.4 \times 10^{-3}$ | 0.85 | 1.04 (1.03–1.06) | $8.8 \times 10^{-7}$ | 0.81 | 1.04 (1.03–1.06) | $5.3 \times 10^{-9}$ | 5.1 | 0.39 |
| rs2758598 | 1 | 156194339 | A | G | 1q22 | 0.16 | 1.07 (1.03–1.11) | $1.8 \times 10^{-4}$ | 0.33 | 1.03 (1.02–1.05) | $8.4 \times 10^{-7}$ | 0.31 | 1.04 (1.02–1.05) | $3.6 \times 10^{-9}$ | 57.7 | 0.01 |
| rs6756513 | 2 | 70172587 | A | G | 2p13.3 | 0.30 | 0.96 (0.94–0.99) | 0.01 | 0.29 | 0.96 (0.95–0.98) | $4.2 \times 10^{-7}$ | 0.29 | 0.96 (0.95–0.98) | $1.5 \times 10^{-8}$ | 0 | 0.80 |
| rs73006998 | 3 | 150464271 | A | G | 3q25.1 | 0.33 | 0.92 (0.89–0.94) | $2.4 \times 10^{-9}$ | 0.03 | 0.94 (0.91–0.98) | $5.8 \times 10^{-3}$ | 0.22 | 0.93 (0.90–0.95) | $1.1 \times 10^{-10}$ | 10.0 | 0.35 |
| rs1281251 | 3 | 156519412 | T | TTGTGAC | 3q25.31 | 0.18 | 0.94 (0.90–0.98) | $1.9 \times 10^{-3}$ | 0.39 | 0.97 (0.96–0.98) | $4.2 \times 10^{-7}$ | 0.37 | 0.97 (0.95–0.98) | $8.4 \times 10^{-9}$ | 24.5 | 0.24 |
| rs11944638 | 4 | 48227719 | T | C | 4p11 | 0.74 | 1.08 (1.04–1.11) | $6.0 \times 10^{-6}$ | 0.93 | 1.05 (1.02–1.08) | $3.1 \times 10^{-4}$ | 0.85 | 1.06 (1.04–1.08) | $1.6 \times 10^{-8}$ | 0 | 0.83 |
| rs11947923 | 4 | 53911337 | T | C | 4q12 | 0.28 | 0.96 (0.93–0.99) | 0.01 | 0.37 | 0.97 (0.96–0.98) | $1.0 \times 10^{-6}$ | 0.36 | 0.97 (0.96–0.98) | $4.5 \times 10^{-8}$ | 0 | 0.76 |
| rs6555134 | 5 | 2776483 | T | C | 5p15.33 | 0.26 | 0.95 (0.92–0.98) | $1.5 \times 10^{-3}$ | 0.58 | 0.97 (0.96–0.98) | $3.6 \times 10^{-7}$ | 0.54 | 0.97 (0.95–0.98) | $2.9 \times 10^{-9}$ | 0 | 0.77 |
| rs7765429 | 6 | 21904169 | T | C | 6p22.3 | 0.89 | 0.94 (0.90–0.98) | $6.8 \times 10^{-3}$ | 0.46 | 0.97 (0.96–0.98) | $3.3 \times 10^{-7}$ | 0.49 | 0.97 (0.96–0.98) | $1.7 \times 10^{-8}$ | 6.4 | 0.38 |
| rs7768862 | 6 | 85088846 | A | T | 6q14.3 | 0.29 | 0.95 (0.92–0.97) | $1.7 \times 10^{-4}$ | 0.51 | 0.97 (0.96–0.98) | $6.4 \times 10^{-6}$ | 0.48 | 0.97 (0.96–0.98) | $2.0 \times 10^{-8}$ | 0.0 | 0.52 |
| rs6940159 | 6 | 170332621 | A | C | 6q27 | 0.82 | 0.94 (0.91–0.97) | $4.6 \times 10^{-4}$ | 0.38 | 0.97 (0.96–0.98) | $2.7 \times 10^{-7}$ | 0.43 | 0.96 (0.95–0.98) | $1.7 \times 10^{-9}$ | 7.7 | 0.37 |
| rs144145984 | 8 | 23644003 | CT | C | 8p21.2 | 0.43 | 0.96 (0.94–0.99) | $3.4 \times 10^{-3}$ | 0.57 | 0.97 (0.96–0.98) | $1.7 \times 10^{-6}$ | 0.55 | 0.97 (0.96–0.98) | $2.4 \times 10^{-8}$ | 0 | 0.56 |
| rs2849506 | 8 | 101329134 | C | G | 8q22.2 | 0.49 | 0.96 (0.93–0.98) | $9.9 \times 10^{-4}$ | 0.40 | 0.97 (0.96–0.98) | $7.5 \times 10^{-6}$ | 0.41 | 0.97 (0.96–0.98) | $4.7 \times 10^{-8}$ | 0 | 0.94 |
| rs142360995 | 8 | 118205719 | A | G | 8q24.11 | 0.09 | 1.13 (1.07–1.18) | $4.1 \times 10^{-6}$ | 0.20 | 1.03 (1.02–1.05) | $1.0 \times 10^{-5}$ | 0.19 | 1.04 (1.03–1.06) | $3.0 \times 10^{-8}$ | 64.0 | 0.003 |
| rs10820600 | 9 | 106856692 | T | C | 9q31.1 | 0.82 | 0.95 (0.92–0.99) | $7.6 \times 10^{-3}$ | 0.44 | 0.97 (0.96–0.98) | $1.8 \times 10^{-7}$ | 0.48 | 0.97 (0.96–0.98) | $5.7 \times 10^{-9}$ | 29.1 | 0.19 |
| rs541079479 | 10 | 22861533 | CA | C | 10p12.2 | 0.13 | 1.06 (1.01–1.11) | 0.01 | 0.42 | 1.03 (1.02–1.05) | $7.0 \times 10^{-7}$ | 0.39 | 1.03 (1.02–1.05) | $4.9 \times 10^{-8}$ | 0 | 0.83 |
| rs2901157 | 10 | 119262365 | A | G | 10q26.11 | 0.75 | 1.06 (1.03–1.09) | $4.2 \times 10^{-4}$ | 0.89 | 1.05 (1.02–1.07) | $2.3 \times 10^{-6}$ | 0.85 | 1.05 (1.03–1.07) | $4.0 \times 10^{-9}$ | 0 | 1 |
| rs10838267 | 11 | 44368892 | A | G | 11p11.2 | 0.33 | 1.06 (1.03–1.09) | $8.2 \times 10^{-5}$ | 0.54 | 1.03 (1.02–1.05) | $3.2 \times 10^{-7}$ | 0.51 | 1.04 (1.03–1.05) | $4.2 \times 10^{-10}$ | 11.9 | 0.34 |
| rs78588049 | 12 | 69180907 | A | ATTTT | 12q15 | 0.15 | 0.93 (0.90–0.97) | $7.5 \times 10^{-4}$ | 0.20 | 0.96 (0.95–0.98) | $3.3 \times 10^{-6}$ | 0.19 | 0.96 (0.95–0.97) | $3.0 \times 10^{-8}$ | 4.0 | 0.40 |
| rs855596 | 12 | 103045519 | T | C | 12q23.2 | 0.07 | 0.90 (0.86–0.95) | $8.3 \times 10^{-5}$ | 0.03 | 0.92 (0.89–0.96) | $1.9 \times 10^{-5}$ | 0.04 | 0.91 (0.89–0.94) | $7.5 \times 10^{-9}$ | 5.0 | 0.39 |
| rs9316500 | 13 | 51094114 | T | G | 13q14.3 | 0.36 | 1.05 (1.02–1.08) | $4.0 \times 10^{-4}$ | 0.71 | 1.03 (1.02–1.05) | $6.7 \times 10^{-6}$ | 0.64 | 1.03 (1.02–1.05) | $2.1 \times 10^{-8}$ | 5.7 | 0.39 |
| rs75004998 | 14 | 77517786 | A | G | 14q24.3 | 0.51 | 0.96 (0.94–0.99) | $7.8 \times 10^{-3}$ | 0.33 | 0.97 (0.96–0.98) | $1.8 \times 10^{-6}$ | 0.36 | 0.97 (0.96–0.98) | $4.9 \times 10^{-8}$ | 0 | 0.92 |
| rs8027365 | 15 | 75808740 | A | C | 15q24.2 | 0.62 | 1.05 (1.02–1.08) | $1.3 \times 10^{-3}$ | 0.73 | 1.04 (1.02–1.05) | $9.7 \times 10^{-8}$ | 0.71 | 1.04 (1.03–1.05) | $4.6 \times 10^{-10}$ | 8.4 | 0.37 |
| rs76535198 | 16 | 71892498 | A | C | 16q22.2 | 0.72 | 1.08 (1.04–1.11) | $1.2 \times 10^{-6}$ | 0.86 | 1.04 (1.03–1.06) | $2.3 \times 10^{-6}$ | 0.83 | 1.05 (1.04–1.07) | $5.4 \times 10^{-11}$ | 0.7 | 0.43 |
| rs12481286 | 20 | 52287610 | T | G | 20q13.2 | 0.31 | 1.05 (1.01–1.08) | $3.5 \times 10^{-3}$ | 0.24 | 1.04 (1.03–1.06) | $1.0 \times 10^{-7}$ | 0.26 | 1.04 (1.03–1.06) | $1.1 \times 10^{-9}$ | 0 | 0.52 |
| rs35418111 | 21 | 47856670 | A | G | 21q11.21 | 0.20 | 1.07 (1.04–1.11) | $3.2 \times 10^{-5}$ | 0.07 | 1.06 (1.04–1.09) | $6.1 \times 10^{-7}$ | 0.12 | 1.07 (1.05–1.09) | $1.1 \times 10^{-10}$ | 0 | 0.97 |
| rs34331122 | 22 | 19762428 | CTT | C | 22q11.21 | 0.56 | 0.94 (0.91–0.97) | $3.7 \times 10^{-5}$ | 0.46 | 0.97 (0.96–0.98) | $7.2 \times 10^{-6}$ | 0.47 | 0.97 (0.96–0.98) | $1.0 \times 10^{-8}$ | 2.2 | 0.41 |

BP base position, NCBI build 37, EAF effect allele frequency, OR odds ratio, CI confidence interval.

**Table 2 Association analysis of 28 newly associated SNPs by estrogen receptor status.**

| SNP | Chr | BP | Effect | Other | ER positive | | | ER negative | | | $I^2$, % | $P_{het}$ |
|---|---|---|---|---|---|---|---|---|---|---|---|---|
| | | | | | EAF | OR (95% CI) | P | EAF | OR (95% CI) | P | | |
| rs72906468 | 1 | 17772093 | A | T | 0.76 | 1.03 (1.02–1.05) | $6.9 \times 10^{-5}$ | 0.75 | 1.04 (1.01–1.06) | $2.0 \times 10^{-3}$ | 0 | 0.75 |
| rs3790585 | 1 | 46023356 | A | T | 0.81 | 1.05 (1.03–1.06) | $7.3 \times 10^{-7}$ | 0.80 | 1.03 (1.00–1.05) | 0.05 | 28.8 | 0.24 |
| rs2758598 | 1 | 156194339 | A | G | 0.32 | 1.03 (1.01–1.05) | $3.6 \times 10^{-5}$ | 0.31 | 1.02 (1.00–1.04) | 0.10 | 0 | 0.37 |
| rs6756513 | 2 | 70172587 | A | G | 0.29 | 0.97 (0.95–0.98) | $6.9 \times 10^{-6}$ | 0.29 | 0.98 (0.96–1.00) | 0.12 | 33.4 | 0.22 |
| rs73006998 | 3 | 150464271 | A | G | 0.20 | 0.91 (0.88–0.93) | $3.6 \times 10^{-10}$ | 0.24 | 0.96 (0.92–1.00) | 0.07 | 81.5 | 0.02 |
| rs11281251 | 3 | 156519412 | T | TTGTGAC | 0.37 | 0.96 (0.95–0.98) | $1.7 \times 10^{-7}$ | 0.36 | 0.96 (0.94–-0.98) | $3.9 \times 10^{-4}$ | 24.5 | 0.24 |
| rs11944638 | 4 | 48227719 | T | C | 0.88 | 1.07 (1.04–1.09) | $8.5 \times 10^{-7}$ | 0.86 | 1.03 (1.00–1.07) | 0.07 | 45.8 | 0.17 |
| rs11947923 | 4 | 53911337 | T | C | 0.36 | 0.97 (0.95–0.98) | $2.4 \times 10^{-6}$ | 0.36 | 0.96 (0.94–0.98) | $4.5 \times 10^{-4}$ | 0 | 0.79 |
| rs6555134 | 5 | 2776483 | T | C | 0.55 | 0.96 (0.95–0.98) | $1.4 \times 10^{-7}$ | 0.53 | 0.97 (0.95–0.99) | $8.4 \times 10^{-3}$ | 0 | 0.45 |
| rs7765429 | 6 | 21904169 | T | C | 0.49 | 0.96 (0.94–0.97) | $8.8 \times 10^{-10}$ | 0.50 | 1.00 (0.98–1.02) | 0.79 | 90.1 | 0.002 |
| rs7768862 | 6 | 85088846 | A | T | 0.48 | 0.97 (0.96–0.98) | $1.6 \times 10^{-5}$ | 0.47 | 0.97 (0.95–0.99) | $2.6 \times 10^{-3}$ | 0 | 0.92 |
| rs6940159 | 6 | 170332621 | T | C | 0.42 | 0.97 (0.95–0.98) | $1.8 \times 10^{-6}$ | 0.44 | 0.97 (0.95–1.00) | 0.02 | 0 | 0.49 |
| rs144145984 | 8 | 23644003 | CT | C | 0.55 | 0.96 (0.95–0.97) | $1.3 \times 10^{-8}$ | 0.54 | 1.00 (0.97–1.02) | 0.65 | 86.9 | 0.006 |
| rs2849506 | 8 | 101329134 | C | G | 0.41 | 0.97 (0.95–0.98) | $1.5 \times 10^{-6}$ | 0.42 | 0.99 (0.97–1.01) | 0.15 | 55.5 | 0.13 |
| rs142360995 | 8 | 118205719 | A | G | 0.20 | 1.04 (1.02–1.06) | $4.0 \times 10^{-6}$ | 0.19 | 1.04 (1.01–1.06) | $7.4 \times 10^{-3}$ | 0 | 0.72 |
| rs10820600 | 9 | 106856692 | T | C | 0.48 | 0.97 (0.96–0.99) | $2.9 \times 10^{-4}$ | 0.49 | 0.96 (0.94–0.98) | $4.5 \times 10^{-4}$ | 0 | 0.36 |
| rs541079479 | 10 | 22861533 | CA | C | 0.40 | 1.04 (1.02–1.05) | $1.0 \times 10^{-6}$ | 0.38 | 1.03 (1.00–1.05) | 0.02 | 0 | 0.45 |
| rs2901157 | 10 | 119262365 | A | G | 0.86 | 1.05 (1.02–1.07) | $2.8 \times 10^{-5}$ | 0.85 | 1.05 (1.02–1.08) | $1.5 \times 10^{-3}$ | 0 | 0.81 |
| rs10838267 | 11 | 44368892 | A | G | 0.52 | 1.03 (1.02–1.05) | $9.4 \times 10^{-6}$ | 0.51 | 1.04 (1.01–1.06) | $7.9 \times 10^{-4}$ | 0 | 0.75 |
| rs78588049 | 12 | 69180907 | A | ATTTT | 0.19 | 0.95 (0.93–0.97) | $3.1 \times 10^{-9}$ | 0.19 | 0.98 (0.96–1.01) | 0.21 | 79.7 | 0.03 |
| rs855596 | 12 | 103045519 | T | C | 0.04 | 0.92 (0.88–0.95) | $3.9 \times 10^{-6}$ | 0.05 | 0.93 (0.88–0.98) | $5.4 \times 10^{-3}$ | 0 | 0.74 |
| rs9316500 | 13 | 51094114 | T | G | 0.65 | 1.03 (1.02–1.05) | $2.4 \times 10^{-5}$ | 0.63 | 1.02 (1.00–1.04) | 0.11 | 8.3 | 0.30 |
| rs75004998 | 14 | 77517786 | A | G | 0.36 | 0.97 (0.96–0.98) | $2.2 \times 10^{-5}$ | 0.37 | 0.97 (0.95–0.99) | $3.2 \times 10^{-3}$ | 0 | 0.96 |
| rs8027365 | 15 | 75808740 | A | C | 0.71 | 1.04 (1.02–1.05) | $8.0 \times 10^{-7}$ | 0.71 | 1.05 (1.03–1.08) | $9.9 \times 10^{-6}$ | 0 | 0.38 |
| rs76535198 | 16 | 71892498 | A | C | 0.83 | 1.05 (1.03–1.07) | $3.1 \times 10^{-6}$ | 0.83 | 1.06 (1.03–1.09) | $8.3 \times 10^{-5}$ | 0 | 0.55 |
| rs12481286 | 20 | 52287610 | T | G | 0.25 | 1.06 (1.04–1.07) | $6.9 \times 10^{-11}$ | 0.26 | 1.02 (0.99–1.04) | 0.20 | 85.1 | 0.01 |
| rs35418111 | 21 | 47856670 | A | G | 0.11 | 1.07 (1.04–1.09) | $6.8 \times 10^{-8}$ | 0.12 | 1.05 (1.02–1.09) | $3.6 \times 10^{-3}$ | 0 | 0.48 |
| rs34331122 | 22 | 19762428 | CTT | C | 0.47 | 0.96 (0.95–0.97) | $1.7 \times 10^{-8}$ | 0.48 | 0.98 (0.96–1.00) | 0.06 | 59.2 | 0.12 |

*BP* base position, NCBI build 37, *EAF* effect allele frequency, *OR* odds ratio, *CI* confidence interval.

**Associations of previously reported risk variants in Asians**. Of the 183 risk variants of breast cancer reported previously, 11 and 172 were originally discovered in studies conducted in Asians and European-ancestry populations, respectively. We were able to investigate 166 variants because 15 variants originally discovered in European populations were (nearly) monomorphic in Asians and two in high LD with rs2747652 (*ESR1*, 6q25.1) were removed. Of the 166 SNPs, 78 were found to be associated with breast cancer risk at $P < 0.05$, while 131 showed associations that were consistent in direction with those originally reported (Supplementary Data 5). Associations for five variants achieved genome-wide significance ($P < 5 \times 10^{-8}$, Asians), with two at 6q25.1 (*ESR1* and *TAB2*), and one each at 15q26.1 (*PRC1*), 16q12.1 (*TOX3*), and 21q22.12 (*LINC00160*). Additionally, borderline genome-wide significant associations were found in seven loci including 2q14.1, 2q35, 3p24.1, 5q33.3, 9q33.3, 12p13.1 and 17q22 ($P < 1 \times 10^{-6}$ in Asians).

**Independent association signals within known susceptibility loci**. We searched extensively for additional independent associations in Asians by conducting conditional analysis for variants located 500 kb of the 166 previously reported SNPs. A total of 820 SNPs from 21 loci were associated with breast cancer risk after conditioning on known risk variants in Asians (Supplementary Data 6). Eight loci, 5q11.2, 6q25.1, 9p21.3, 10q21.2, 12q24.21, 16q12.1, 18q12.3 and 21q21.1, may harbor independent association signals with genome-wide significance (Table 3, conditional $P < 5 \times 10^{-8}$ in Asians). Five of these eight loci, including 5q11.2, 9p21.3, 12q24.21, 18q12.3, and 21q21.1, have not previously been linked to breast cancer risk in Asian populations. Significant heterogeneity between Asian and European-ancestry populations was observed ($P_{heterogeneity} < 0.05$) at 5q11.2, 9p21.3, 12q24.21, 16q12.1, and 21q21.1, and the strength of the association was stronger in Asian than European-ancestry women.

**Polygenic risk scores**. We evaluated the association between PRS and breast cancer risk among SWHS participants, a subset of samples included in the Asia Breast Cancer Consortium. The PRS was generated using the weights (βs) obtained from Asian-specific meta-analysis. Women with a high estimated PRS had a 3.6-fold higher risk of breast cancer compared to those who had a low PRS (highest decile vs. lowest decile, Supplementary Table 10).

## Discussion

This large-scale meta-analysis, including approximately 310,000 women of Asian and European ancestry and represents the largest GWAS to identify genetic determinants for breast cancer. In addition to identifying 31 potential novel risk loci for breast cancer (Table 1, Supplementary Table 8, and Statistical Methods), we replicated in Asian women 78 of the GWAS-identified risk variants for breast cancer. Since the risk variants initially reported in European populations might not be the lead SNPs in Asians, we performed further analyses to show that 21 known susceptibility loci may harbor additional independent signals, of which 16 showed at least one stronger association than the originally reported risk SNP. Our study has generated substantial novel information to improve the understanding of breast cancer genetics and etiology and provides clues for future studies to functionally characterize the risk variants and candidate genes identified in our study.

Similar to other GWAS, nearly all of the newly identified risk variants mapped to intergenic regions or introns of genes. One

**Table 3 Eight novel breast cancer risk-associated SNPs located within previously known loci in Asians: a conditional analysis.**

| SNP | Chr | BP | Effect | Other | Reported | Locus | Nearest gene | EAF | OR (95% CI) | P | $I^2$, % | $P_{het}$ |
|---|---|---|---|---|---|---|---|---|---|---|---|---|
| rs112776581 | 5 | 56054333 | T | TA | rs62355902 | 5q11.2 | LOC105378979 | 0.11 | 1.21 (1.15–1.27) | $3.5 \times 10^{-14}$ | 0 | 0.70 |
| rs2941741 | 6 | 152008982 | A | G | rs9397437, rs2747652 | 6q25.1 | ESR1 | 0.13 | 1.13 (1.08–1.17) | $8.2 \times 10^{-10}$ | 0 | 0.62 |
| rs974336 | 9 | 22006348 | T | C | rs1011970 | 9p21.3 | CDKN2B | 0.22 | 1.10 (1.06–1.13) | $5.9 \times 10^{-9}$ | 24.6 | 0.22 |
| rs78053936 | 10 | 64300331 | A | C | rs10822013, rs10995201 | 10q21.2 | ZNF365 | 0.80 | 1.11 (1.07–1.15) | $1.7 \times 10^{-8}$ | 20.4 | 0.27 |
| rs61929345 | 12 | 116001403 | T | G | rs1292011 | 12q24.21 | LOC105370003 | 0.16 | 1.11 (1.07–1.15) | $4.9 \times 10^{-8}$ | 8.8 | 0.36 |
| rs3803661 | 16 | 52586477 | A | G | rs4784227 | 16q12.1 | CASC16 | 0.63 | 1.08 (1.05–1.12) | $3.7 \times 10^{-8}$ | 0 | 0.61 |
| rs12455117 | 18 | 42884026 | A | T | rs6507583 | 18q12.3 | SLC14A2 | 0.68 | 1.09 (1.06–1.12) | $1.7 \times 10^{-8}$ | 0 | 0.74 |
| rs2823126 | 21 | 16561704 | A | G | rs2823093 | 21q21.1 | NRIP1 | 0.28 | 0.90 (0.88–0.93) | $1.1 \times 10^{-10}$ | 39.5 | 0.12 |

*BP base position, NCBI build 37, EAF effect allele frequency, OR odds ratio, CI confidence interval.*

exception was rs10820600, which is located in the 5′-UTR region of the *SMC2* gene. *SMC2* encodes the structural maintenance of chromosomes protein-2, an essential subunit of the condensin complex I and II. The protein is critically involved in chromosome condensation and segregation during cell cycles[17]. Emerging evidence shows that *SMC2* mutations and dysregulated expression are associated with multiple cancers[18].

Of the thirteen lead risk variants located in the introns of genes, six showed strong evidence of cis-regulation for seven genes nearby, including *YBEY, SNUPN, MAN2C1, LINC00886, TBX1, SEMA4A, and MUTYH*. For example, the locus at 21q22.3 (rs35418111) showed compelling evidence of influencing expression of *YBEY*, a gene that encodes a highly conserved metalloprotein. Our gene-based analysis indicated a potential oncogenic role of *YBEY* in breast cancer development. Although the function of *YBEY* has not been fully elucidated, dysregulation of its expressions caused by copy number variation has been found in familial and early-onset breast cancer[19], as well as colorectal cancer[20]. Further, we showed that *MAN2C1* may play a protective role against breast carcinogenesis in the gene-based analysis. However, another study found that *MAN2C1* promotes cancer growth via a negative regulation of phosphatase and tensin homolog (*PTEN*) function in prostate and breast cancer cell lines[21]. These results suggested that *MAN2C1* may have distinct functional impact on cancer initiation compared to that on tumor progression. Few studies have investigated the mechanistic roles of *LINC00886, SNUPN* and *SEMA4A* in cancer initiation. Germline mutations in *SEMA4A* have been linked to the predisposition of familial colorectal cancer type X[22]. Our study provides the first evidence linking these two genes to breast cancer susceptibility.

Potential candidate genes were also revealed by the newly associated variants lying in the intergenic regions between coding genes. *LOXL2* and *STC1* were pinpointed as cis targets of rs144145984 at 8p21.2. *LOXL2* is a member of the lysyl oxidase family of amine oxidases and *STC1* belongs to the glycoprotein hormones family. Research regarding the functions of *LOXL2* and *STC1* in cancer development is limited. However, pre-clinical studies have implicated *LOXL2* and *STC1* in the progression of breast cancer[23,24]. Inhibiting *LOXL2* activity shows a 55–75% decrease in primary tumor volume in female athymic nude mice, which were implanted with MDA-MB-231 human breast cancer cells[23]. The reduction in tumor burden was suspected to be mediated by the inhibition of angiogenesis. A recent study suggested the role of *STC1* played in the breast tumorigenesis could be subtype-dependent[24]. A cancer promoting function was found in murine mammary tumor cells and human triple negative breast cancer lines (MDA-MB-231), while an opposite function

was shown in luminal breast cancer lines (ER+/PR+, T47D cells).

The pleiotropy of rs855596 at 12q23.2 provided a plausible mechanistic link for the observed genetic association with breast cancer risk. The minor (T) allele of rs855596 is associated with decreased breast cancer risk and is linked to the minor allele G of the nearby rs703556 ($r^2 = 0.94$ in EA and 0.43 in East Asians). The G allele of rs703556 is associated with lower mammographic dense area in women[25]. Mammographic density, an established risk factor for breast cancer[26], is a measure based on the radiographic appearance of the breast by mammography. Several loci were related to other cancers or benign tumors. SNPs in 22q11.21, 1q22 and 4q12 were found to be associated with risk of prostate cancer[27], testicular germ cell tumor[28] and leiomyoma, respectively[29]. We hypothesize potential underlying mechanisms via hormone metabolism for these loci. Variants at 10p12.2 (*PIP4K2A*) indicated an association with risk of acute lymphoblastic leukemia[30] and 6p22.3 (*CASC15*) with endometrial cancer[31], lung cancer[32], and neuroblastoma[33]. These regions implicated in genetic susceptibility across different types of cancers may serve as prioritized target of interest for future fine-mapping studies. For some of the phenotypes like blood cell counts and sodium levels, we currently lack the proper knowledge to decipher the likely mechanisms that link them to breast cancer development.

Notable racial heterogeneity was found for the loci at 1q22 (rs2758598) and 8q24.11 (rs142360995), which may reflect the differential regional LD structures and allele frequency between the two populations at these loci. The effect sizes in Asians are larger than those in European populations for both SNPs, over four times for rs142360995 and two times for rs2758598. The association at 3q25.1 (rs73006998) was dominant by estimates in Asians (ABCC: $2.4 \times 10^{-9}$; in BCAC, $P = 5.8 \times 10^{-3}$), although no heterogeneity was observed. Previously, the same locus was reported to be associated with hormonal receptor-positive breast cancer, with a borderline genome-wide significance in a Japanese population (rs6788895, LD $r^2 = 0.76$ in East Asians)[34]. We found significant heterogeneity by ER status for this locus and the association was primarily driven by ER-positive cancer. Racial heterogeneity was also observed for many known risk variants initially reported in European populations. It may be attributable to multiple factors including the Winner's curse[35], population-specific LD structure, and false positives in the original GWAS.

Sixty-seven of the 155 index SNPs originally reported in European-ancestry women were replicated in women of Asian descent at $P < 0.05$. For those not replicated in our analysis, possible explanations include differences in local LD structure and genetic architecture for the disease between these two

populations and a relatively small sample size of Asian studies. In summary, in this large GWAS including 147,183 breast cancer cases and 130,749 unaffected controls, we identified 31 potential novel breast cancer susceptibility loci by meta-analyzing data of two large consortia conducted in Asian and European women. Using an independent set of 16,787 cases and 16,680 controls of Asian ancestry, we evaluated 22 lead variants and found that all variants showed the same direction of the association, although only ten of them were statistically significant. As many of the associations were driven by GWAS of European women and the sample size of our replication set was small, the low replication rate is not unexpected. Nevertheless, our study reveals many novel loci and potential targeted genes that may influence breast cancer susceptibility, although the possibility of false-positives for some loci cannot be completely ruled out. Future investigations are warranted to replicate our findings.

## Methods

**Study population.** The overall cross-ancestry meta-analysis was conducted using data from two large consortia, the ABCC and BCAC. Detailed descriptions of participating studies are included in Supplementary Note 1. Briefly, in the ABCC, genome-wide SNP data were generated from 24,206 breast cancer cases and 24,775 unaffected controls recruited from studies conducted in mainland China, South Korea, and Japan (Supplementary Table 1). The BCAC-Asian dataset was composed of COGS ($N = 10,716$) and OncoArray projects ($N = 14,337$); twelve studies contributed samples to either or both projects. The BCAC-European dataset consisted of three sub-sets, GWAS ($N = 32,498$), COGS ($N = 89,677$), and OncoArray projects ($N = 106,776$)[8]. A total of 80,428 and 26,948 cases had ER-positive and -negative breast cancer, respectively.

Included as a replication set were an additional 10,829 cases and 10,996 controls of Asian ancestry, recruited by eight studies from South Korea, Japan, Hong Kong, and Taiwan (Supplementary Note 1). There was no overlap in samples from participating studies.

**Genotyping and quality control.** All of the genotyping and quality control procedures for GWAS, except for the expanded MEGA$^{EX}$ chip, have been described elsewhere[3,4,6–12,34,36,37] (Supplementary Table 1). The MEGA$^{EX}$ chip contains approximately 2.04 million variants with an excellent genomic coverage of common variants (a minor allele frequency of 0.01 or higher) across multi-racial populations. We added to the MEGA$^{EX}$ chip ~80k variants selected from our GWAS of breast and colorectal cancers and exome sequencing data for breast cancer cases in Asian-ancestry populations. In total, 2.1 million variants were included on this array. Quality control (QC) procedure include: samples were excluded if they (i) had genotyping call rate <95%; (ii) were male based on genotype data; (ii) had a close relationship with a Pi-HAT estimate >0.25; (iii) were heterozygosity outliers; (iv) were ancestry outliers. SNPs were excluded if they had (i) a call rate <95%; (ii) no clear genotyping clusters; (iii) a minor allele frequency <0.001; (iv) a Hardy-Weinberg equilibrium test of $P < 1 \times 10^{-6}$; (v) genotyping concordance < 95% among the duplicated QC samples[3,4,6–12,34,36,37]. All of the datasets were imputed using the 1000 Genomes Project Phase 3 mixed populations as the reference panel, except for the BioBank Japan (BBJ1) study, in which the HapMap Phase II (release 22) was used. Only SNPs with an imputation $R^2 > 0.3$ were included in the further analyses.

Genotyping of the replication set of cases and controls was completed using the iPLEX Sequenom MassArray platform (Agena Bioscience Inc., San Diego, California, USA). One negative control (water), two blinded duplicates and two samples from the HapMap project were included as QC samples in each 96-well plate. Samples or SNPs that had a genotyping call rate of <95% were excluded. We also excluded SNPs that had a concordance with the QC samples of <95% or an unclear genotype call. If the assay could not be designed for the lead SNP, a surrogate SNP which is in LD with the lead SNP with $r^2 > 0.8$ in Asians (1000 Genome) was selected. Of the 28 newly identified risk variants, 22 were successfully genotyped by Sequenom and evaluated in the association analysis, while six failed in the probe designing stage. Additional 11,642 independent samples from MYBRCA and SGBCC studies (Supplementary Note 1) were also included in the replication stage in evaluation of the 22 newly identified risk variants.

**Statistical methods.** Logistic regression analysis was performed within each study of Asian women to obtain a per-allele odds ratio (OR) for each SNP using PLINK2.0[38]. Principal components analyses were conducted within each GWAS dataset. Age and the top two PCs were included as covariates for in all regression models. Study (COGS) or country/region (OncoArray) was also included in the analyses of BCAC data[8]. The number of PCs to be included in the regression was determined by evaluation of Scree plot. Sensitivity analyses were conducted to include top 10 PCs, which showed very similar ORs as those derived from analyses adjusted for two PCs (Supplementary Table 11). A meta-analysis was performed

using METAL[39] with a fixed-effects model to generate Asian-specific and cross-ancestry estimates. Heterogeneity was assessed by the Cochran's Q statistic and I$^2$. For the cross-ancestry meta-analysis, we were mainly interested in evaluating variants that were associated with breast cancer risk at $P < 0.01$ in the Asian-specific analysis ($n_{snp} = 244,746$). However, three additional lead SNPs that did not meet this criterion can also be found in Supplementary Table 8. One representative SNP with the lowest p value was reported as the index SNP for each of the newly identified loci after variant pruning (LD $r^2 < 0.1$). The significant locus is considered novel if it is located 500 kb away from the 183 known risk loci for breast cancer The LD with known risk SNPs was also checked to verify the independence. Among the newly associated loci, we further applied the method implemented in MR-MEGA[40] to account for the population heterogeneity for two loci showing significant heterogeneity in the cross-ancestry fixed-effect meta-analysis. The results were shown in the Supplementary Table 9. The association was slightly more significant than the original fixed-effect meta-analysis for these two loci. Inflation of the test statistics ($\lambda$) was estimated by dividing the 50th percentile of the test statistic by 0.455 (the 50th percentile for a $\chi^2$ distribution on 1 degree of freedom)[41]. We standardized the inflation statistic to account for the large size of our study by calculating $\lambda_{1000}$ ($\lambda$ for an equivalent study with 1000 cases and 1000 controls)[8]. For the replication stage, analyses were conducted with an adjustment for age and study.

For each of the Asian studies with GWAS data (Supplementary Table 1), we searched for independent secondary association signals within a flanking +/− 500 kb region of the lead variant in each of the newly identified breast cancer risk loci using conditional analysis, with an adjustment for the newly identified lead risk SNPs when individual-level data was available $\left[\log\left(\frac{P}{1-P}\right) = \beta_0 + \beta_1 SNP_i + \beta_2 SNP_{new\,index} + \beta COVAR\right]$. We used GCTA software (option -COJO)[42] to perform the conditional analysis for the BBJ1, Seoul Breast Cancer Study (SeBCS), and BCAC European GWAS, for which only summary statistics data were available. MEGA array genotyping data was used as reference panel for LD estimation. The results of individual study were combined by a fixed-effect meta-analysis using METAL. SNPs showing an association with breast cancer risk at $P_{conditional} < 1 \times 10^{-4}$ were considered independent secondary association signals. The analysis was also performed within known susceptibility loci. All statistical tests were two-sided.

**Statistical power.** For the cross-ancestry meta-analysis (sample size shown in the Supplementary Table 1, alpha set to $5.0 \times 10^{-8}$), we had >80% power to detect the association between SNP and breast cancer risk with an OR of >1.06, 1.07, and 1.11 and EAF of 0.10 in the analysis of ER-positive, ER-negative cancer and all cancer combined, respectively (Supplementary Table 18).

**Functional annotation and enrichment analysis.** Novel risk loci were defined as those ±500 Kb away from the lead risk variant reported by previous GWAS conducted in populations of Asian or European-ancestry for breast cancer. The lead risk SNPs newly identified in our study were defined as the variant showing an association with breast cancer risk with the lowest *P*-value in a given locus in the meta-analysis. Functional annotations of the lead SNPs and their correlated variants ($r^2 > 0.8$ in 1000 Genomes Project, East Asian or European populations) were performed using HaploReg v4.1[43]. The functional annotation of chromatin states from chromHMM, DNase I hypersensitive and histone modifications such as H3K4, H3K9 and H3K27, were based on the epigenetic data in human breast mammary epithelial cells (HMEC), MCF-7 cells, and other cell lines from the Encyclopedia of DNA Elements (ENCODE) Project and Roadmap Epigenetics Project. We further applied GARFIELD[44] to assess functional enrichment for all risk loci identified to date for breast cancer risk and those newly reported in the current study. According to GARFIELD, the significance level for the enrichment analysis was set to $9.7 \times 10^{-5}$. Known risk loci (±500 kb) were removed when evaluating functional enrichment for the newly identified loci.

**Expression quantitative loci (eQTL) analysis.** To identify target genes, we performed eQTL analysis utilized four independent sets of gene expression data derived from normal breast ($N = 85$, GTEx, women of European ancestry), breast tumor (women of European ancestry, TCGA, $N = 672$; METABRIC, $N = 1904$) and adjacent normal tissues (women of Asian ancestry, SBCGS, $N = 151$). We focused on *cis*-eQTL analyses for genes residing ±500 Kb flanking each newly associated leading SNP. The details of data processing were described in Supplementary Note 2.

A linear regression model was used to perform eQTL analyses to estimate the additive effect for each leading SNP identified on gene expression levels. We additionally adjusted for somatic copy number alteration and methylation levels in the regression model for the analysis of TCGA data. We only adjusted for somatic copy number alteration in the analysis for the METABRIC set.

**Gene-based analysis.** We recently conducted a transcriptome wide association study (TWAS) to investigate associations of genetically predicted gene expression with the risk of breast cancer[45]. We utilized the same approach to examine the associations with breast cancer risk of genes located within flanking 500 kb of each newly associated leading SNP. The breast-specific prediction model was generated

using the elastic net method as implemented in the glmnet R package ($\alpha = 0.5$), with tenfold cross-validation[45]. To further increase statistical power, we also utilized 6,124 samples across 39 tissue types from 369 unique European individuals who had genome-wide genotype data available to build cross-tissue models[46,47]. The expression of a gene for individual $i$ in tissue $t$, $Y_{i,t}$, is modeled as $Y_{i,t} = Y_i^{CT} + Z_i'\beta + \varepsilon_{i,t}$, where $Y_i^{CT}$ represents the cross-tissue component of expression levels for a given gene. The mixed effect model parameters were estimated using the lme4 package in R. The predicted gene expressions $\widehat{Y}_i$ in the breast-specific models and $\widehat{Y_i^{CT}}$ in the cross-tissue models then were evaluated for their associations with breast cancer risk in the ABCC and BCAC, using methods implemented in MetaXcan[48].

**Polygenic risk score**. We used the 11 risk SNPs originally reported in Asian populations, 28 newly identified SNPs from the current analysis (Table 1), and 28 risk SNPs originally identified in European populations that were replicated in the Asian populations in this current study (Supplementary Data 5, $P < 0.05/166$) to generate polygenetic risk score (PRS). PRS were calculated as $PRS = \sum \beta_i SNP_i$. The weights, $\beta$s, used to generate the score were obtained from Asian-specific meta-analysis. The association between the score and breast cancer risk was tested in the samples from Shanghai Women's Health Study (SWHS, $N$ total = 2427, $N$ case = 368, $N$ control = 2059), which were also contributed to the Asian MEGA project. The PRS was tested in both continuous (1 SD change) and categorical forms (deciles in controls). The area under the curve was also calculated to show its discriminatory ability. Overfitting is less a concern as SWHS participants only accounted for a very small proportion in the Asian-specific meta-analysis (~8%).

**Reporting summary**. Further information on research design is available in the Nature Research Reporting Summary linked to this article.

## Data availability

Access to the ABCC data could be requested by submission of an inquiry to Dr. Wei Zheng (wei.zheng@vanderbilt.edu). Request of access to the BCAC data could be submitted directly to BCAC (http://bcac.ccge.medschl.cam.ac.uk/). Access to other data: GTEx: https://gtexportal.org/home/datasets; TCGA - https://portal.gdc.cancer.gov/; METABRIC: https://www.ebi.ac.uk/ega/studies/EGAS00000000083.

## Code availability

Access to the custom code could be requested by submission of an inquiry to Dr. Wei Zheng (wei.zheng@vanderbilt.edu).

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

## Acknowledgements

The content is solely the responsibility of the authors and does not necessarily represent the official views of the funding agents. The funders had no role in study design, data collection and analysis, decision to publish, or preparation of the manuscript. This research was supported in part by the US National Institutes of Health grants R01CA124558, R01CA148667, R01CA158473, R01CA064277, R37CA070867, and UM1CA182910 (to W.Z.); R01CA118229 and R01CA092585 (to X.-O.S.); R01CA122756 (to Q.C.); and R01CA137013 (to J. Long), Department of Defense Idea Awards BC011118 (to X.-O.S.) and BC050791 (to Q.C.), and Ingram and Anne Potter Wilson Professorship and Research Reward funds (to W.Z.). Sample preparation and genotyping assays at Vanderbilt were conducted at the Survey and Biospecimen Shared Resources and Vanderbilt Microarray Shared Resource, which are supported in part by the Vanderbilt-Ingram Cancer Center (P30CA068485). Data analyses were conducted using the Advanced Computing Center for Research and Education (ACCRE) at Vanderbilt University. The SeBCS was supported by the BRL (Basic Research Laboratory) program through the National Research Foundation of Korea funded by the Ministry of Education, Science and Technology (2011-0001564). KOHBRA/KOGES was supported by a grant from the National R&D Program for Cancer Control, Ministry for Health, Welfare and Family Affairs, Republic of Korea (#1020350). Studies conducted among Asian women include (Principal Investigator, grant support): the Shanghai Breast Cancer Study (W.Z. and X.-O.S., R01CA064277), the Shanghai Women's Health Study (W.Z., R37CA070867 and UM1CA182910), the Shanghai Breast Cancer Survival Study (X.-O. S., R01CA118229), the Shanghai Endometrial Cancer Study (X.-O.S., R01CA092585, controls only), the Seoul Breast Cancer Study [D.K., BRL (Basic Research Laboratory) program through the National Research Foundation of Korea funded by the Ministry of Education, Science and Technology (2012-0000347)], the BioBank Japan Project (S.-K.L., the Ministry of Education, Culture, Sports, Sciences and Technology from the Japanese Government); the Hwasun Cancer Epidemiology Study-Breast (S.-S.K., the Biobank of Chonnam National University Hwasun Hospital, a member of the Korea Biobank Network, # 07SA2014020), the Nagano Breast Cancer Study (M.I., Grants-in-Aid for the Third Term Comprehensive Ten-Year Strategy for Cancer Control from the Ministry of Health, Labor and Welfare of Japan, and for Scientific Research on Priority Areas, 17015049 and for Scientific Research on Innovative Areas, 221S0001, from the Ministry of Education, Culture, Sports, Science, and Technology of Japan), the Hospital-based Epidemiologic Research Program at Aichi Cancer Center [Grant-in-Aid for Scientific Research on Priority Areas of Cancer (No. 17015018) from the Japanese Ministry of Education, Culture, Sports, Science and Technology and the "Practical Research for Innovative Cancer Control (15ck0106177h0001)" from the Japan Agency for Medical Research and development, AMED (K. Matsuo), and Cancer Bio Bank Aichi; the Asia Cancer Program (K. Muir and A.L., the NIHR Manchester Biomedical Research Centre and by the ICEP and CRUK, # C18281/A19169); the Canadian Breast Cancer Study (K.A. and J. Spinelli, the Canadian Cancer Society, # 313404); the Los Angeles County Asian-American Breast Cancer Case-Control Study (A.H.W., the California Breast Cancer Research Program [1RB-0287, 3PB-0102, 5PB-0018, 10PB-0098]. Incident breast cancer cases were collected by the USC Cancer Surveillance Program (CSP) which is supported under subcontract by the California Department of Health. The CSP is also part of the National Cancer Institute's Division of Cancer Prevention and Control Surveillance, Epidemiology, and End Results Program, under contract number N01CN25403); the Malaysian Breast Cancer Genetic Study (S.-H.T., the Malaysian Ministry of Higher Education [UM.C/HlR/MOHE/06] and Cancer Research Malaysia. MYMAMMO is supported by research grants from Yayasan Sime Darby LPGA Tournament and Malaysian Ministry of Higher Education [RP046B-15HTM]); the Northern California Breast Cancer Family Registry (E.M.J., the National Cancer Institute [USA, UM1 CA164920]. The content of this manuscript does not necessarily reflect the views or policies of the National Cancer Institute or any of the collaborating centers in the Breast Cancer Family Registry (BCFR), nor does mention of trade names, commercial products, or organizations imply endorsement by the USA Government or the BCFR.); the Singapore Breast Cancer Cohort (M.H., the NUS start-up Grant, National University Cancer Institute Singapore [NCIS] Centre Grant and the NMRC Clinician Scientist Award. Additional controls were recruited by the Singapore Consortium of Cohort Studies-Multi-ethnic cohort [SCCS-MEC], which was funded by the Biomedical Research Council, grant number: 05/1/21/19/425); and the Taiwanese Breast Cancer Study (C.-Y.S., the Taiwan Biobank project of the Institute of Biomedical Sciences, Academia Sinica, Taiwan). Studies conducted among European-ancestry women Genotyping of the OncoArray was principally funded by three sources: the PERSPECTIVE project, funded from the Government of Canada through Genome Canada and the Canadian Institutes of Health Research, the Ministère de l'Économie, de la Science et de l'Innovation du Québec through Genome Québec, and the Quebec Breast Cancer Foundation; the NCI Genetic Associations and Mechanisms in Oncology (GAME-ON) initiative and Discovery, Biology and Risk of Inherited Variants in Breast Cancer (DRIVE) project [NIH Grants U19 CA148065, X01HG007492]; and Cancer Research UK [C1287/A10118, C1287/A16563]. The BCAC is funded by Cancer Research UK [C1287/A16563], the European Community's Seventh Framework Programme under grant agreement 223175 [HEALTH-F2-2009-223175] (COGS).

## Author contributions

Study design: W.Z.; Data analysis: X.S., X.G., Y.Y.; Data interpretation: X.S., Jirong Long, Q.C., X.G., Y.Y., J.S., B.L., R.T., X.-O.S., W.Z.; Writing of the manuscript: X.S., W.Z.; Review of the manuscript: X.S., Jirong Long, Q.C., S.-S.K., J.-Y.C., M.K., S.K.P., M.K.B., J.D., Q.W., Y.Y., J.S., X.G., B.L., R.T., K.J.A., K.Y.K.C., T.L.C., Y.-T.G., M.H., W.K.H., Hidemi Ito, M.I., Hiroji Iwata, E.M.J., Y.K., U.S.K., M.-K.K., S.-Y.K., A.W.K., A.K., E.-S.L., Jingmei Li, A.L., S.-K.L., S.M., Koichi Matsuda, Keitaro Matsuo, Kenneth Muir, D.-Y.N., B.P., M.-H.P., C.-Y.S., M.-H.S., J.J.S., A.T., C.T., S.T., A.H.W., Y.-B.X., T.Y., Y.Z., R.L.M., A.M.D., P.D.P.P., M.G.-C., S.-H.T., X.-o.S., D.K., D.F.E., J.S., W.Z.

## Competing Interests
The authors declare no competing interests.

## Additional information

Xiang Shu[1], Jirong Long[1], Qiuyin Cai[1], Sun-Seog Kweon[2,3], Ji-Yeob Choi[4,5,6], Michiaki Kubo[7], Sue K. Park[4,5,6], Manjeet K. Bolla[8], Joe Dennis[8], Qin Wang[8], Yaohua Yang[1], Jiajun Shi[1], Xingyi Guo[1], Bingshan Li[9], Ran Tao[10,11], Kristan J. Aronson[12], Kelvin Y.K. Chan[13,14], Tsun L. Chan[15,16], Yu-Tang Gao[17], Mikael Hartman[18,19,20], Weang Kee Ho[21], Hidemi Ito[22,23], Motoki Iwasaki[24], Hiroji Iwata[25], Esther M. John[26,27,28], Yoshio Kasuga[29], Ui Soon Khoo[13], Mi-Kyung Kim[30], Sun-Young Kong[31,32,33], Allison W. Kurian[27], Ava Kwong[15,34,35], Eun-Sook Lee[31,32,33], Jingmei Li[20,36,37], Artitaya Lophatananon[38,39], Siew-Kee Low[7], Shivaani Mariapun[40], Koichi Matsuda[41], Keitaro Matsuo[42,43], Kenneth Muir[38,39], Dong-Young Noh[6,44], Boyoung Park[45], Min-Ho Park[46], Chen-Yang Shen[47,48], Min-Ho Shin[2], John J. Spinelli[49,50], Atsushi Takahashi[7,51], Chiuchen Tseng[52], Shoichiro Tsugane[53], Anna H. Wu[52], Yong-Bing Xiang[17], Taiki Yamaji[24], Ying Zheng[54], Roger L. Milne[55,56,57], Alison M. Dunning[58], Paul D.P. Pharoah[8,58], Montserrat García-Closas[59], Soo-Hwang Teo[60,61], Xiao-ou Shu[1], Daehee Kang[5,6,62,63], Douglas F. Easton[8,58], Jacques Simard[64] & Wei Zheng[1✉]

[1]Division of Epidemiology, Department of Medicine, Vanderbilt Epidemiology Center, Vanderbilt-Ingram Cancer Center, Vanderbilt University Medical Center, Nashville, TN, USA. [2]Department of Preventive Medicine, Chonnam National University Medical School, Hwasun, Korea. [3]Jeonnam Regional Cancer Center, Chonnam National University Hwasun Hospital, Hwasun, Korea. [4]Department of Biomedical Sciences, Seoul National University College of Medicine, Seoul, Korea. [5]Department of Preventive Medicine, Seoul National University College of Medicine, Seoul, Korea. [6]Cancer Research Institute, Seoul National University College of Medicine, Seoul, Korea. [7]RIKEN Center for Integrative Medical Sciences, Yokohama, Japan. [8]Centre for Cancer Genetic Epidemiology, Department of Public Health and Primary Care, University of Cambridge, Cambridge, UK. [9]Department of Molecular Physiology & Biophysics, Vanderbilt Genetics Institute, Vanderbilt University, Nashville, TN, USA. [10]Department of Biostatistics, Vanderbilt University Medical Center, Nashville, TN, USA. [11]Vanderbilt Genetics Institute, Vanderbilt University Medical Center, Nashville, TN, USA. [12]Department of Public Health Sciences and Queen's Cancer Research Institute, Queen's University, Kingston, ON, Canada. [13]Department of Pathology, Li Ka Shing Faculty of Medicine, University of Hong Kong, Hong Kong SAR, China. [14]Department of Obstetrics & Gynaecology, Li Ka Shing Faculty of Medicine, University of Hong Kong, Hong Kong SAR, China. [15]Hong Kong Hereditary Breast Cancer Family Registry, Hong Kong SAR, China. [16]Department of Molecular Pathology, Hong Kong Sanatorium & Hospital, Hong Kong SAR, China. [17]State Key Laboratory of Oncogene and Related Genes & Department of Epidemiology, Shanghai Cancer Institute, Renji Hospital, Shanghai Jiaotong University School of Medicine, Shanghai, China. [18]Department of Surgery, National University Hospital, Singapore, Singapore. [19]Saw Swee Hock School of Public Health, National University of Singapore, Singapore, Singapore. [20]Department of Surgery, Yong Loo Lin School of Medicine, National University of Singapore, Singapore, Singapore. [21]Department of Applied Mathematics, Faculty of Engineering, University of Nottingham Malaysia Campus, Semenyih, Selangor, Malaysia. [22]Division of Cancer Information and Control, Aichi Cancer Center Research Institute, Nagoya, Japan. [23]Department of Descriptive Cancer Epidemiology, Nagoya University Graduate School of Medicine, Nagoya, Japan. [24]Division of Epidemiology, Center for Public Health Sciences, National Cancer Center, Tokyo, Japan. [25]Department of Breast Oncology, Aichi Cancer Center, Nagoya, Aichi, Japan. [26]Department of Epidemiology, Cancer Prevention Institute of California, Fremont, CA, USA. [27]Departments of Health Research and Policy, School of Medicine, Stanford University, California, CA, USA. [28]Stanford Cancer Institute, Stanford University School of Medicine, California, CA, USA. [29]Department of Surgery, Nagano Matsushiro General Hospital, Nagano, Japan. [30]Division of Cancer Epidemiology and Management, National Cancer Center, Goyang, Korea. [31]National Cancer Center Graduate School of Cancer Science and Policy, Goyang, Republic of Korea. [32]Hospital, National Cancer Center, Goyang, Republic of Korea. [33]Research Institute, National Cancer Center, Goyang, Republic of Korea. [34]Department of Surgery, University of Hong Kong, Hong Kong SAR, China. [35]Department of Surgery, Hong Kong Sanatorium & Hospital, Hong Kong SAR, China. [36]Human Genetics, Genome Institute of Singapore, Singapore, Singapore. [37]Department of Medical Epidemiology and Biostatistics, Karolinska Institutet, Stockholm, Sweden. [38]Division of Health Sciences, Warwick Medical School, Warwick University, Coventry, UK. [39]Institute of Population Health, University of Manchester, Manchester, UK. [40]Cancer Research Malaysia, Subang Jaya, Selangor, Malaysia. [41]Laboratory of Clinical Genome Sequencing, Graduate School of Frontier Sciences, University of Tokyo, Tokyo, Japan. [42]Division of Cancer Epidemiology and Prevention, Aichi Cancer Center Research Institute, Nagoya, Japan. [43]Division of Cancer Epidemiology, Nagoya University Graduate School of Medicine, Nagoya, Japan. [44]Department of Surgery, Seoul National University Hospital, Seoul, South Korea. [45]Department of Medicine, Hanyang University College of Medicine, Seoul, Korea. [46]Department of Surgery, Chonnam National University Medical School, Seoul, Korea. [47]College of Public Health, China Medical University, Taichong, Taiwan. [48]Taiwan Biobank, Institute of Biomedical Sciences, Academia Sinica, Taipei, Taiwan. [49]Population Oncology, BC Cancer, Vancouver, BC, Canada. [50]School of Population and Public Health, University of British Columbia, Vancouver, BC, Canada. [51]Department of Genomic Medicine, Research Institute, National Cerebral and Cardiovascular Center, Suita, Osaka, Japan. [52]Department of Preventive Medicine, Keck School of Medicine, University of Southern California, Los Angeles, CA, USA. [53]Center for Public Health Sciences, National Cancer Center, Tokyo, Japan. [54]Shanghai Municipal Center for Disease Control and Prevention, Shanghai, China. [55]Cancer Epidemiology Division, Cancer Council Victoria, Melbourne, Australia. [56]Centre for Epidemiology and Biostatistics, Melbourne School of Population and Global Health, University of Melbourne, Parkville, Victoria, Australia. [57]Precision Medicine, School of Clinical Sciences at Monash Health, Monash University, Clayton, Victoria, Australia. [58]Centre for Cancer Genetic Epidemiology, Department of Oncology, University of Cambridge, Cambridge, UK. [59]Division of Cancer Epidemiology and Genetics, National Cancer Institute, Bethesda, MD, USA. [60]Cancer Research Malaysia, Subang Jaya, Selangor, Malaysia. [61]Department of Surgery, Faculty of Medicine, University Malaya, Kuala Lumpar, Malaysia. [62]Department of Biomedical Sciences, Seoul National University Graduate School, Seoul, Korea. [63]Institute of Environmental Medicine, Seoul National University Medical Research Center, Seoul, Korea. [64]Genomics Center, Centre Hospitalier Universitaire de Québec - Université Laval, Research Center, Québec City, QC, Canada. ✉email: wei.zheng@vanderbilt.edu

