## [Peer Review File · Nature Communications]

Reviewers' comments:

Reviewer #1 (Remarks to the Author):

Shu et al. have analyzed genome-wide association study (GWAS) data from the Asia Breast Cancer Consortium (ABCC) consisting of 24,206 breast cancer cases and 24,775 controls. They report 28 novel loci associated with the risk of breast cancer in Asians meeting the criteria of genome wide significance. An additional three loci were identified from a meta analysis of both Asian and European women from the Breast Cancer Association Consortium (BCAC) consisting of 122,977 cases and 105,974 controls.

Strengths of this report are the large databases with Asian GWAS data. The statistical analyses are appropriate and thorough. Additional follow-up allowed for replication of 10 loci in an independent sample 16,787 cases and 16,690 controls of Asian ancestry. They also were able to replicate 78 previously reported risk variants in women of Asian descent. Further follow-up to identify target genes from eQTL and TWAS analyses show possible involvement of 9 genes from eight of the 28 novel loci, providing further evidence supporting these associations. Bioinformatics tools also provide evidence in support of the biologic function with many of the SNPs indicative of promoters or enhancers. Additionally, many European ancestry genes did not replicate in Asians.

Since breast cancer is heterogeneous disease with subtype showing differences in susceptibility it is important to determine the associations within the subtypes. The authors determine the genetic association for ER-positive and ER-negative breast cancers and report nominally significant associations for 28 SNPs for ER positive and 14 for ER negative with some significant difference between subtypes. There are no sample sizes provided for the subtype analysis and these should be described in the methods section. It would also be informative to provide the statistical power for these disease subsets.

Other than in silico evidence, support for biological function was not pursued and though the findings provide a foundation for future investigation, this report falls short of in-depth functional characterization of risk variants and genes. Therefore, the significance of these findings are that they provide insight into the genetic susceptibility of breast cancer in Asian women demonstrating there are novel susceptibility loci as well as common loci with women of European Ancestry.

Minor note: There is sentence fragment on page 14 lines 381-382.

Reviewer #2 (Remarks to the Author):

The authors report a transethnic GWAS meta-analysis of breast cancer identifying 31 new loci.

I must say I found the paper very difficult to follow and the presentation needs work on it. Specific issues:

1. A flow diagram in the main text summarising the study design would aid the reader.
2. Figure 1a, the Manhattan plot has not been annotated and is totally useless. It should be deleted. Adds nothing.
3. Figure 2b, the cartoon of the 28 associations across the different chromosomes adds nothing. It should be deleted. They are pointless.
4. Tables 1-3 do not provide sufficient data. The association statistics, heterogeneity P-values should be presented. The word "Test" should be replaced by "Risk" allele. It should be stated what the BP abbreviation refers to and what NCBI build it is referenced to.
5. To me the methodology for performing the meta-analysis could be improved upon. In the absence of individual level data adopting a formal transethnic approach as per MR mega (Morris

Nature Comms) or such like to adjust for ethnic differences.

6. Similarly the precise method used for the conditional analysis is not explained at all. There are formal methods for dealing with this trans-ethnicity. Was this done on each individual series and if so how combined.

7. The threshold for imputation of 0.3 is rather liberal. It should be stated which associations are based on non-directly typed SNPs. For those which based in imputation the fidelity of the imputation should be formally assessed by direct genotyping.

8. The eQTL analysis is simply hand waving picking out eQTLs with an P-value of 0.05. Formal SMR or such should be applied. Similarly formal enrichment analysis should be performed to the histone makers. Simply stating some SNPs maps to a region of predicted enhanced is basically irrelevant.

9. A table detailing the association P-values for all published risk loci should be presented in an easy to digest format. Ideally the false positive association reporting probabilities. If associations are not now significant it should be stated why not.

10. Some commentary work on a PRS would be appropriate.

11. Statement about data access.

Reviewer #3 (Remarks to the Author):

This is a nicely written paper presenting results from breast cancer GWAS among Asian women. The authors detected and replicated a number of associated SNPs, and undertook eQTL analyses to further evaluate these findings. The key novelty here is the meta-analysis of samples from Asian GWAS, undertaken to increase the power for detecting new results. I have a few specific comments.

1. The eQTL analysis was undertaken by TWAS imputation into four breast tissue databases. Some of these tissue are normal breast, and others are tumor. The implications of using both normal and cancerous tissue for the eQTL analyses should be considered. What does it mean to have germline imputation with tumor expression?

2. It is unclear whether linkage disequilibrium was considered for the identification of novel SNPs. COJO was used to identify additional secondary signals, but it seems the only criteria for the novel 28 SNPs was that they were >500Kb away from any of the known 183 loci. The investigators should also include LD filtering against the known loci to assess independent novelty.

3. Why were only 2 principal components adjusted for? Was there no variation beyond this? The conventional approach is more on the order of 10 PCs, and the sensitivity of the findings to using so few for adjustment should be assessed.

4. The pleiotropic associations (Suppl Table 4) are not discussed (beyond mammographic density). Are these relevant phenotypes, risk factors?

5. For the replication of the 28 novel hits, 22 were looked up in an independent sample of Asian women and only 10 had $p < 0.05$. The other 18 were replicated in European Ancestry women. If the focus of this paper is on identifying Asian-specific risk SNPs, then shouldn't the focus be on the 10 unique SNPs? Does the lack of replication of the other 18 in the Asian population indicate that they are primarily relevant to Europeans, and not Asians?

Authors' Responses to Reviewers' Comments

Reviewer #1 (Remarks to the Author):

Shu et al. have analyzed genome-wide association study (GWAS) data from the Asia Breast Cancer Consortium (ABCC) consisting of 24,206 breast cancer cases and 24,775 controls. They report 28 novel loci associated with the risk of breast cancer in Asians meeting the criteria of genome wide significance. An additional three loci were identified from a meta analysis of both Asian and European women from the Breast Cancer Association Consortium (BCAC) consisting of 122,977 cases and 105,974 controls.

Strengths of this report are the large databases with Asian GWAS data. The statistical analyses are appropriate and thorough. Additional follow-up allowed for replication of 10 loci in an independent sample 16,787 cases and 16,690 controls of Asian ancestry. They also were able to replicate 78 previously reported risk variants in women of Asian descent. Further follow-up to identify target genes from eQTL and TWAS analyses show possible involvement of 9 genes from eight of the 28 novel loci, providing further evidence supporting these associations. Bioinformatics tools also provide evidence in support of the biologic function with many of the SNPs indicative of promoters or enhancers. Additionally, many European ancestry genes did not replicate in Asians.

1. Since breast cancer is heterogeneous disease with subtype showing differences in susceptibility it is important to determine the associations within the subtypes. The authors determine the genetic association for ER-positive and ER-negative breast cancers and report nominally significant associations for 28 SNPs for ER positive and 14 for ER negative with some significant difference between subtypes. There are no samples sizes provided for the subtype analysis and these should be described in the methods section. It would also be informative to provide the statistical power for these disease subsets.

Response: We appreciate the positive feedback from the reviewer. A total of 80,428 and 26,948 patients with ER-positive and ER negative breast cancer were included in the stratified analysis, respectively. The statistical power for the stratified analysis is now included in the Methods section. The table below shows the statistical power for various scenarios. The population risk of breast cancer is set to 130.8/100,000 (ACS: [cancer.org/content/dam/cancer-org/research/cancer-facts-and-statistics/breast-cancer-facts-and-figures/breast-cancer-facts-and-figures-2019-2020.pdf](https://www.cancer.org/content/dam/cancer-org/research/cancer-facts-and-statistics/breast-cancer-facts-and-figures/breast-cancer-facts-and-figures-2019-2020.pdf)). The α level is set to 5×10^{-8} . The power calculation is based on the sample size of 277,932 (overall breast cancer), 20,6105 (ER-positive), and 152,625 (ER-negative).

Overall breast cancer, $N_{cases} = 147,183, N_{control} = 130,749$		
Effect allele frequency	Effect size (OR)	Power
0.05	1.08	0.815
0.10	1.06	0.870
0.20	1.05	0.969
0.25	1.05	0.993
ER-positive breast cancer, $N_{cases} = 80,428, N_{control} = 125,677$		
0.05	1.10	0.886
0.10	1.07	0.841
0.20	1.06	0.972
0.25	1.06	0.994
ER-negative breast cancer, $N_{cases} = 26,948, N_{control} = 125,677$		
0.05	1.15	0.888
0.10	1.11	0.905
0.20	1.08	0.870
0.25	1.08	0.951

Methods:

“A total of 80,428 and 26,948 cases had ER-positive and -negative breast cancer, respectively.”

“Statistical power

For the cross-ancestry meta-analysis (sample size shown in the Supplementary Table 1, alpha set to 5.0×10^{-8}), we had > 80% power to detect the association between SNP and breast cancer risk with an OR of > 1.06, 1.07, and 1.11 and EAF of 0.10 in the

analysis of ER-positive, ER-negative cancer and all cancer combined, respectively (Supplementary Table 18)."

2. Other than in silico evidence, support for biological function was not pursued and though the findings provide a foundation for future investigation, this report falls short of in-depth functional characterization of risk variants and genes. Therefore, the significance of these findings are that they provide insight into the genetic susceptibility of breast cancer in Asian women demonstrating there are novel susceptibility loci as well a common loci with women of European Ancestry.

Response: As pointed out by the reviewer, the goal of our study is to identify novel susceptibility loci for breast cancer in both women of Asian and European ancestry. We agree with the reviewer that it is important to perform functional characterization of risk variants and genes in the future. Carrying out these functional assays is a very significant undertaking and thus is beyond the scope of this study. Our study will provide important clues for other research groups to investigate functional significance of the identified risk variants and genes in the future.

Minor note: There is sentence fragment on page 14 lines 381-382.

Response: We have made the correction accordingly.

"Samples or SNPs that had a genotyping call rate of < 95% were excluded."

Reviewer #2 (Remarks to the Author):

1. A flow diagram in the main text summarising the study design would aid the reader.

Response: We have added a new Supplementary Figure 1 to summarize the design of this study.

2. Figure 1a, the Manhattan plot has not been annotated and is totally useless. It should be deleted. Adds nothing.

Response: We removed the original Figure 1a.

3. Figure 2b, the cartoon of the 28 associations across the different chromosomes adds nothing. It should be deleted. They are pointless.

Response: We removed the original Figure 1b.

4. Tables 1-3 do not provide sufficient data. The association statistics, heterogeneity P-values should be presented. The word "Test" should be replaced by "Risk" allele. It should be stated what the BP abbreviation refers to and what NCBI build it is referenced to.

Response: We have made the changes. The association estimates (OR and 95% CI) are presented in the Table 1-3. Both I^2 and heterogeneity P-values are also presented for the overall cross-ancestry meta-analysis, stratified analysis by ER status, and conditional analysis. The “Test allele” is now replaced by “Effect allele” since not all these alleles were associated with an elevated risk. We have now changed “BP” to “Base Position, NCBI build 37”. For each newly associated locus, we also provided detailed estimates for its association and heterogeneity test in each individual Asian study (the Supplementary Table 2 in the original submission).

5. To me the methodology for performing the meta-analysis could be improved upon. In the absence of individual level data adopting a formal transethnic approach as per MR mega (Morris Nature Comms) or such like to adjust for ethnic differences.

Response: As suggested, we conducted the cross-ancestry meta-analysis using the method implemented in the MR-MEGA. Two of the 31 newly associated loci were significantly heterogeneous between Asian and European descendants ($P_{het} < 0.05$, 1q22-rs2758598 and 8q24.11-rs142360995). This method was able to derive a smaller p value.

We add the paragraph below to the Methods sections to describe the procedure and findings.

Methods:

“Among the newly associated loci, we further applied the method implemented in MR-MEGA to account for the population heterogeneity for two loci showing significant heterogeneity in the cross-ancestry fixed-effect meta-analysis. The results were shown in the Supplementary Table 15. The association was slightly more significant than the original fixed-effect meta-analysis for these two loci.”

6. Similarly the precise method used for the conditional analysis is not explained at all. There are formal methods for dealing with this trans-ethnicity. Was this done on each individual series and if so how combined.

Response: We added the following paragraph to clarify the procedures that how conditional analysis was conducted.

“For each of the Asian studies with GWAS data (Supplementary Table 1), we searched for independent secondary association signals within a flanking +/- 500kb region of the lead variant in each of the newly identified breast cancer risk loci using conditional analysis, with an adjustment for the newly identified lead risk SNPs when individual-level data was available [$\log\left(\frac{P}{1-P}\right) = \beta_0 + \beta_1 SNP_i + \beta_2 SNP_{new\ index} + \beta COVAR$]. We used GCTA software (option -COJO)1 to perform the conditional analysis for the BBJ1, Seoul Breast Cancer Study (SeBCS), and BCAC European GWAS, for which only summary statistics data were available. MEGA array genotyping data was used as reference panel for LD estimation. The results of individual study were combined by a fixed-effect meta-analysis using METAL. SNPs showing an association with breast

cancer risk at P conditional $< 1 \times 10^{-4}$ were considered independent secondary association signals. The analysis was also performed within known susceptibility loci.”

7. The threshold for imputation of 0.3 is rather liberal. It should be stated which associations are based on non-directly typed SNPs. For those which based in imputation the fidelity of the imputation should be formally assessed by direct genotyping.

Response: The detailed information of imputation quality for each newly associated SNP was presented in the original Supplementary Table 2 for every individual Asian study. On average, the imputation quality is high (Mean=0.93, Median=0.97) for the newly associated SNPs. We directly genotyped a group of newly identified SNPs in 1,324 repeated samples. The consistent rates are very high (see below).

Concordance between direct genotyping and imputed data for selected newly associated SNPs

SNP	Concordant rate
rs3739737 ^a	1.00
rs10838267	0.998
rs35418111	0.968
rs6756513	0.984
rs73006998	0.998
rs75004998	0.987
rs7768862	1.00
rs2849506	0.99
rs11947923	1.00
rs4886712 ^a	0.998
rs6555134	0.982
rs6940159	0.995
rs855596	0.997
rs9316500	1.00
rs78588049	0.986
rs34331122	0.983

^a Proxy SNPs were identified and genotyped for rs10820600 (rs3739737, LD=0.99) and rs201439118 and (rs4886712, LD=1) in East Asian populations in 1000 Genome Project.

8. The eQTL analysis is simply hand waving picking out eQTLs with a P-value of 0.05. Formal SMR or such should be applied. Similarly, formal enrichment analysis should be performed to the histone makers. Simply stating some SNPs maps to a region of predicted enhanced is basically irrelevant.

Response: The eQTL analysis served the first step to characterize the potential relationship between risk variants, gene expression, and cancer risk. To be conservative, we utilized multiple datasets to first identify promising cis-eQTLs for the newly identified risk variants. We then performed a gene-based analysis adopting the

same approach used in transcriptome-wide association study (TWAS), i.e. PrediXcan, to formally test the association between genetically predicted gene expression and breast cancer risk for these loci. The underlying concept is virtually identical to the SMR, although there are some technical differences. Our gene-based analysis can improve statistical power since it incorporates information from multiple cis-eQTL variants instead of focusing on one SNP at a time in the eQTL and SMR analyses.

As suggested, we performed formal analyses to assess enrichment of functional annotations including histone marks and chromatin states for the newly associated regions. The procedure and findings were described in the Methods and Results sections.

Methods:

“We further applied GARFIELD to assess functional enrichment for all risk loci identified to date for breast cancer risk and those newly reported in the current study. According to GARFIELD, the significance level for the enrichment analysis was set to 9.7×10^{-5} . Known risk loci (± 500 kb) were removed when evaluating functional enrichment for the newly identified loci.”

Results:

“The enrichment analysis supported this observation (Supplementary Figure 2-A). Of particular note is a strikingly strong enrichment signal of transcribed chromatin states that was found for the newly associated loci when compared to all risk loci (Supplementary Figure 2-B). Enrichment signals of multiple histone modifications were also observed for both newly identified and overall association loci (Supplementary Figure 2-C & D). The newly identified loci were enriched particularly for H4K78me2 and H4K20me1. These results indicated that the newly identified loci are tightly involved in active gene transcription events.”

9. A table detailing the association P-values for all published risk loci should be presented in an easy to digest format. Ideally the false positive association reporting probabilities. If associations are not now significant it should be stated why not.

Response: The association and P value were reported in the Supplementary Table 12 in the original submission. All 11 SNPs originally discovered by GWAS in the population of Asian ancestry were replicated. Sixty-seven of the 155 previously reported risk variants among European populations were replicated at $P < 0.05$ in women of Asian ancestry. Among the 155 SNPs, associations for 129 were consistent in direction. We further provided FPRP in the table. We also added a few sentences to the Discussion to explain why some risk variants identified in European descendants are not replicated in Asians.

“Sixty-seven of the 155 index SNPs originally reported in European-ancestry women were replicated in women of Asian descent at $P < 0.05$. For those not replicated in our analysis, possible explanations include differences in local LD structure and genetic architecture for the disease between these two populations and a relatively small sample size of Asian studies.”

10. Some commentary work on a PRS would be appropriate.

Response: As suggested, we generated a polygenetic risk score. The weights used to generate the score were obtained from Asian-specific meta-analysis and cross-ancestry meta-analysis. The paragraphs below describe how we generated the score and the corresponding findings in the Methods and Results, respectively.

Methods

“Polygenic risk scores

We used the 11 risk SNPs originally reported in Asian populations, 28 newly identified SNPs from the current analysis, and 28 risk SNPs originally identified in European populations that were replicated in the Asian populations in this current study ($P < 0.05/166$) to generate polygenetic risk score (PRS). PRS were calculated as $PRS = \sum \beta_i SNP_i$. The weights, β_s , used to generate the score were obtained from Asian-specific meta-analysis. The association between the score and breast cancer risk was tested in the samples from Shanghai Women’s Health Study (SWHS, $N_{total} = 2,427$, $N_{case} = 368$, $N_{control} = 2,059$), which were also contributed to the Asian MEGA project. The PRS was tested in both continuous (1 SD change) and categorical forms (deciles in controls). The area under the curve was also calculated to show its discriminatory ability. Overfitting is less a concern as SWHS participants only accounted for a very small proportion in the Asian-specific meta-analysis (~8%).”

Results

“Polygenic risk scores

We evaluated the association between PRS and breast cancer risk among SWHS participants, a subset of samples included in the Asia Breast Cancer Consortium. The PRS was generated using the weights (β_s) obtained from Asian-specific meta-analysis. Women with a high estimated PRS had a 3.6-fold higher risk of breast cancer compared to those who had a low PRS (highest decile vs. lowest decile, Supplementary Table 16).”

11. Statement about data access.

Response: We have now added the following statement to describe data access in the manuscript.

“Access to the ABCC data could be requested by submission of data sharing inquiry to Dr. Wei Zheng (wei.zheng@vanderbilt.edu). Request of access to the BCAC data could be submitted directly to BCAC (<http://bcac.ccge.medschl.cam.ac.uk/>). Access to other data: GTEx: <https://gtexportal.org/home/datasets>; TCGA - <https://portal.gdc.cancer.gov/>; METABRIC: <https://www.ebi.ac.uk/ega/studies/EGAS00000000083>.

Reviewer #3 (Remarks to the Author):

This is a nicely written paper presenting results from breast cancer GWAS among Asian women. The authors detected and replicated a number of associated SNPs, and undertook eQTL analyses to further evaluate these findings. The key novelty here is the meta-analysis of samples from Asian GWAS, undertaken to increase the power for detecting new results. I have a few specific comments.

1. The eQTL analysis was undertaken by TWAS imputation into four breast tissue databases. Some of these tissue are normal breast, and others are tumor. The implications of using both normal and cancerous tissue for the eQTL analyses should be considered. What does it mean to have germline imputation with tumor expression?

Response: We first performed eQTL analyses in normal and cancer tissues to identify candidate genes for downstream analyses of the association of eQTL-identified genes with breast cancer risk. Ideally, only normal breast tissues should be included for this purpose, but it was unlikely for us to obtain more normal tissues in addition to those from GTEx. As shown in the previous study conducted by Li et al (Cell, PMID: 23374354), eQTL analysis can also be conducted in the tumor tissues with the proper adjustment of methylation and copy number variation. We used multiple variants to predict the expression of nine genes and then evaluated the association of predicted gene expression with breast cancer risk. The genetic prediction models were built using the data of normal breast tissues in the GTEx project. The details could be found in our previous publication, which was also cited in the manuscript (Wu et al, Nat Genet. PMID: 29915430).

2. It is unclear whether linkage disequilibrium was considered for the identification of novel SNPs. COJO was used to identify additional secondary signals, but it seems the only criteria for the novel 28 SNPs was that they were >500Kb away from any of the known 183 loci. The investigators should also include LD filtering against the known loci to assess independent novelty.

Response: When reporting the newly associated SNPs, we only selected the SNP with the lowest p-value within a LD block, which was also > 500 Kb away from any of the known 183 loci. The LD with known loci in the same chromosome was checked to ensure the independence. We then performed conditional analysis to search for secondary signals within the locus by adjusting for the most significant SNP. Similar procedure was performed for the known 183 loci where previously reported risk SNPs were taken into consideration. The results of conditional analysis for the newly associated and known loci were shown in Supplementary Table 6 and Supplementary Table 13, respectively. We added the description below to clarify how we define novel risk SNP.

“One representative SNP with the lowest p value was reported as the index SNP for each of the newly identified loci after variant pruning ($LD\ r^2 < 0.1$). The significant locus

is considered novel if it is located 500kb away from the 183 known risk loci for breast cancer. The LD with known risk SNPs was also checked to verify the independence.”

3. Why were only 2 principal components adjusted for? Was there no variation beyond this? The conventional approach is more on the order of 10 PCs, and the sensitivity of the findings to using so few for adjustment should be assessed.

Response: The top 2 PCs adjusted in the analysis was determined by evaluation of Scree plot for each study having individual-level data. Adjusting for 10 PCs is not supported by this assessment. Nevertheless, we conducted sensitivity analysis to show that adjustment of top ten PCs virtually did not change the risk estimates for any of the newly identified risk SNPs. We have added the sentence below to the Method section.

“The number of PCs to be included in the regression was determined by evaluation of Scree plot. Sensitivity analyses were conducted to include top 10 PCs, which showed very similar ORs as those derived from analyses adjusted for two PCs (Supplementary Table 17).”

SNP	Original results		New results with 10 PCs adjusted
	EAF	OR (95% CI)	OR (95% CI)
rs72906468	0.76	1.04 (1.03-1.05)	1.04 (1.02-1.05)
rs3790585	0.81	1.04 (1.03-1.06)	1.04 (1.03-1.06)
rs2758598	0.31	1.04 (1.02-1.05)	1.04 (1.02-1.05)
rs6756513	0.29	0.96 (0.95-0.98)	0.97 (0.95-0.98)
rs73006998	0.22	0.93 (0.90-0.95)	0.92 (0.90-0.95)
rs11281251	0.37	0.97 (0.95-0.98)	0.97 (0.95-0.98)
rs11944638	0.85	1.06 (1.04-1.08)	1.06 (1.04-1.08)
rs11947923	0.36	0.97 (0.96-0.98)	0.97 (0.96-0.98)
rs6555134	0.54	0.97 (0.95-0.98)	0.97 (0.95-0.98)
rs7765429	0.49	0.97 (0.96-0.98)	0.97 (0.96-0.98)
rs7768862	0.48	0.97 (0.96-0.98)	0.97 (0.96-0.98)
rs6940159	0.43	0.96 (0.95-0.98)	0.97 (0.96-0.98)

rs144145984	0.55	0.97 (0.96-0.98)	0.97 (0.96-0.98)
rs2849506	0.41	0.97 (0.96-0.98)	0.97 (0.96-0.98)
rs142360995	0.19	1.04 (1.03-1.06)	1.04 (1.03-1.06)
rs10820600	0.48	0.97 (0.96-0.98)	0.97 (0.96-0.98)
rs541079479	0.39	1.03 (1.02-1.05)	1.03 (1.02-1.05)
rs2901157	0.85	1.05 (1.03-1.07)	1.05 (1.03-1.07)
rs10838267	0.51	1.04 (1.03-1.05)	1.04 (1.02-1.05)
rs78588049	0.19	0.96 (0.95-0.97)	0.96 (0.95-0.97)
rs855596	0.04	0.91 (0.89-0.94)	0.91 (0.89-0.94)
rs9316500	0.64	1.03 (1.02-1.05)	1.03 (1.02-1.05)
rs75004998	0.36	0.97 (0.96-0.98)	0.97 (0.96-0.98)
rs8027365	0.71	1.04 (1.03-1.05)	1.04 (1.03-1.05)
rs76535198	0.83	1.05 (1.04-1.07)	1.05 (1.03-1.07)
rs12481286	0.26	1.04 (1.03-1.06)	1.04 (1.03-1.06)
rs35418111	0.12	1.07 (1.05-1.09)	1.07 (1.05-1.09)
rs34331122	0.47	0.97 (0.96-0.98)	0.97 (0.96-0.98)

4. The pleiotropic associations (Suppl Table 4) are not discussed (beyond mammographic density). Are these relevant phenotypes, risk factors?

Response: We intended to keep the manuscript concise. Therefore, we only discussed the pleiotropic associations related to mammographic density (a known risk factor for breast cancer) and other cancer risks. Based on the current knowledge, we lack firm evidence linking some of the phenotypes such as platelet count, hemoglobin concentration, sodium levels etc. to breast cancer development. We added the following statement to describe the uncertainty.

“For some of the phenotypes like blood cell counts and sodium levels, we currently lack the proper knowledge to decipher the likely mechanisms that link them to breast cancer development.”

5. For the replication of the 28 novel hits, 22 were looked up in an independent sample of Asian women and only 10 had $p < 0.05$. The other 18 were replicated in European Ancestry women. If the focus of this paper is on identifying Asian-specific risk SNPs, then shouldn't the focus be on the 10 unique SNPs? Does the lack of replication of the other 18 in the Asian population indicate that they are primarily relevant to Europeans, and not Asians?

Response: The current study aims to identify novel susceptibility loci for breast cancer by combining GWAS data from two racial groups, Asians and Europeans. The findings would be applicable to both populations. The replication set is quite small, which may explain why some of the identified associations were not replicated. As part of a large collaboration initiated recently, additional samples are being assembled to identify new breast cancer loci and replicate previous findings. However, it would take at least three years for this collaboration to start analyzing data.

REVIEWERS' COMMENTS:

Reviewer #1 (Remarks to the Author):

This is a revision of a manuscript of genome-wide associations from the Asia Breast Cancer Consortium (ABCC) that found 28 novel loci associated with the risk of breast cancer in Asians. Additional loci were identified from a meta analysis of both Asian and European women from the Breast Cancer Association Consortium (BCAC). Subtype specific associations are also determined. The findings of this report add to a body of literature of breast cancer GWAS results but uniquely address genetic associations among Asian women.

The authors have satisfactorily addressed the concerns from the initial review. Importantly, they have clarified the power of their samples, size, improved the methodology of the analysis, demonstrated they have appropriately controlled for admixture, and pointed out that they have taken the science as far as is practical at this time. Future research can be leveraged from this findings of this report.

Reviewer #2 (Remarks to the Author):

The revised paper is much improved.

The authors have addressed my concerns.

Reviewer #3 (Remarks to the Author):

Following up on a comment made in the previous review, for the replication of the 28 novel hits, 22 were looked up in an independent sample of Asian women and only 10 had $p < 0.05$. The other 18 were not replicated. It still seems that the 10 replicated SNPs should be the focus of the paper, not the remaining 18 that have not been replicated in an independent population.

Authors' Responses to Reviewers' Comments

Reviewer #1 (Remarks to the Author):

This is a revision of a manuscript of genome-wide associations from the Asia Breast Cancer Consortium (ABCC) that found 28 novel loci associated with the risk of breast cancer in Asians. Additional loci were identified from a meta analysis of both Asian and European women from the Breast Cancer Association Consortium (BCAC). Subtype specific associations are also determined. The findings of this report add to a body of literature of breast cancer GWAS results but uniquely address genetic associations among Asian women.

The authors have satisfactorily addressed the concerns from the initial review. Importantly, they have clarified the power of their samples, size, improved the methodology of the analysis, demonstrated they have appropriately controlled for admixture, and pointed out that they have taken the science as far as is practical at this time. Future research can be leveraged from this findings of this report.

Response: We appreciate the positive feedback from the reviewer.

Reviewer #2 (Remarks to the Author):

The revised paper is much improved.

The authors have addressed my concerns.

Response: We appreciate the positive feedback from the reviewer.

Reviewer #3 (Remarks to the Author):

Following up on a comment made in the previous review, for the replication of the 28 novel hits, 22 were looked up in an independent sample of Asian women and only 10 had $p < 0.05$. The other 18 were not replicated. It still seems that the 10 replicated SNPs should be the focus of the paper, not the remaining 18 that have not been replicated in an independent population.

Response: The findings of 28 novel loci are robust given that the results were derived from the combination of two large consortia. Due to issues in probe design, some SNPs cannot be evaluated in the replication set. Of important note, the association direction was consistent in the replication samples for all the SNPs evaluated although some did not reach the significance threshold (Supplementary Table 5). We have acknowledged the limitations by admitting insufficient sample size for replication set and likelihood of chance finding in the revised manuscript (see below). However, false negatives also have a detrimental impact in scientific research therefore we would like to report all SNPs as potential targets for future research.

Discussion

“As many of the associations were driven by GWAS of European women, the low replication rate is not unexpected. Unfortunately, we could not include a further independent dataset of European ancestry to optimize the power in the replication stage. Nevertheless, our study reveals many novel loci and potential targeted genes that may influence breast cancer susceptibility, although the possibility of false-positives cannot be completely ruled out. Future investigations are warranted to replicate our findings.”